# Non-Destructive Hyperspectral Imaging for Rapid Determination of Catalase Activity and Ageing Visualization of Wheat Stored for Different Durations

**DOI:** 10.3390/molecules27248648

**Published:** 2022-12-07

**Authors:** Yurong Zhang, Guanqiang Lu, Xianqing Zhou, Jun-Hu Cheng

**Affiliations:** 1School of Food and Strategic Reserves, Henan University of Technology, Zhengzhou 450001, China; 2Engineering Research Center of Grain Storage and Security of Ministry of Education, Zhengzhou 450001, China; 3Henan Provincial Engineering Technology Research Center on Grain Post Harvest, Zhengzhou 450001, China; 4School of Food Science and Engineering, South China University of Technology, Guangzhou 510641, China

**Keywords:** wheat, catalase activity, hyperspectral imaging technology, ageing, wavelengths selection, visualization

## Abstract

(1) In order to accurately judge the new maturity of wheat and better serve the collection, storage, processing and utilization of wheat, it is urgent to explore a fast, convenient and non-destructively technology. (2) Methods: Catalase activity (CAT) is an important index to evaluate the ageing of wheat. In this study, hyperspectral imaging technology (850–1700 nm) combined with a BP neural network (BPNN) and a support vector machine (SVM) were used to establish a quantitative prediction model for the CAT of wheat with the classification of the ageing of wheat based on different storage durations. (3) Results: The results showed that the model of 1ST-SVM based on the full-band spectral data had the best prediction performance (R^2^ = 0.9689). The SPA extracted eleven characteristic bands as the optimal wavelengths, and the established model of MSC-SPA-SVM showed the best prediction result with R^2^ = 0.9664. (4) Conclusions: The model of MSC-SPA-SVM was used to visualize the CAT distribution of wheat ageing. In conclusion, hyperspectral imaging technology can be used to determine the CAT content and evaluate wheat ageing, rapidly and non-destructively.

## 1. Introduction

Catalase activity (CAT) is the main source of the elimination of hydrogen peroxide in plants [1]. The plant metabolism process produces the reactive oxygen free radicals, and then transforms them into hydrogen peroxide, which has a damaging effect on the plant cells [2]. CAT is commonly found in plant tissues and cells. It is an anti-ageing protective enzyme for organisms. It is of great significance to protect the stability and integrity of the cell membranes, and it is related to seed vigor [3,4]. It has the functions of improving plant photosynthesis, anti-stress levels, enhancing the defense ability, delaying senescence, promoting metabolism, etc. [5,6]. The study found that the CAT activity of rice grains showed a decreasing trend with the extension of the storage time. CAT is an important indicator for judging the ageing of rice grains [7]. Using visible spectroscopy and near-infrared spectroscopy to quickly analyze the superoxide dismutase (SOD) activity in barley leaves, the predicted correlation coefficient (r) and root mean square error (RMSEP) were 0.9064 and 0.5336, respectively [8]. Near-infrared technology was also used to monitor the biocatalytic reactions and peroxidase (POD) activity [9]. The hyperspectral imaging technology combining traditional spectroscopy and imaging technology to obtain the spectral information and spatial distribution information of the sample at the same time was applied to detect the POD activity of gray disease in tomato leaves [10]. It has the characteristics of “map integration”, and it has been widely used in the field of agricultural products in recent years, which can realize rapid and non-destructive testing [11,12,13,14,15]. As the hyperspectral technology has the advantages of accuracy, rapidity, non-destructiveness, and high efficiency in food inspection, hyperspectral technology has now rapidly developed into a very advantageous analytical technique [16,17,18,19].

Hyperspectral imaging (HSI) has proven to be a very promising technology because the data obtained using this technology contain enough information about the characteristics of the sample. Thus, it can replace the human inspectors or wet chemical methods for the automatic grading and nutrition of food. Hyperspectral data can combine the discriminative analysis techniques (QDA, LDA, and PLS-DA) to explore the qualitative characteristics of the samples or integrate the regression analysis techniques (PCR, PLSR, SVM, ANN, NB, and PLS-DA) to explore quantitative characteristics at the same time [20,21]. The potential of HSI in estimating the concentration of phenolic flavor compounded on malted barley was studied. The correlation coefficients between the predicted values of SWIR and UV and the true values were 0.99 and 0.91, respectively [22]. The application of near-infrared hyperspectral technology in grain identification, quality, moisture, hardness, protein and starch content determination, germination detection and food safety were reviewed (Fox and Manley 2014; Hussain et al., 2019) [12,13]. Many scholars have classified the aging of meat, and they have achieved good results [23,24,25,26].

In recent years, hyperspectral analysis technology has also made quite good achievements in grains. First of all, the hyperspectral analysis technology has achieved great success in the quality analysis of rice [14,27,28,29]. Secondly, this kind of technology showed high accuracy in the detection of aflatoxin on corn kernels, the identification of the varieties, the identification of haploid kernels, and the detection of the water content [30,31,32,33]. In addition, in-depth research and good results have been achieved in wheat pests, virus infections, and grain storage quality analyses [34,35,36,37,38,39].

However, using hyperspectral imaging technology combined with algorithms to predict CAT and classify the different storage durations of wheat has rarely been reported. Thus, this study mainly investigated the CAT changes of wheat during artificially accelerated ageing and applied the hyperspectral imaging technology combined with chemometric methods to predict the wheat CAT quantitatively and to distinguish the wheat ageing degree to realize the rapid and non-destructive evaluation of wheat quality.

## 2. Results and Discussion

### 2.1. Changes in Wheat CAT Activity during Ageing

The change of the wheat CAT activity is shown in Figure 1. Due to a large number of samples, each point in the figure represented the average of the seven samples that were taken. It can be seen from Figure 1 that with the prolonging of the ageing time, the CAT of the strong, medium, and weak gluten wheat all showed different degrees of a downward trend.

The changing trend of CAT was consistent with the report of Feng et al. on the relationship between CAT activity and the changes in the physical wheat quality [40]. Moreover, the changes of CAT in different wheat varieties were the same. Among them, the CAT of weak gluten wheat decreased the fastest, from the initial 117.78 mg H_2_O_2_/g to 53.93 mg H_2_O_2_/g, which is a decrease of 63.85 mg H_2_O_2_/g. The CAT decrease of the medium gluten wheat was the second fastest, from the initial 112.98 mg H_2_O_2_/g 67.56 mg H_2_O_2_/g, which is a decrease of 45.42 mg H_2_O_2_/g. The strong gluten wheat CAT decreased the slowest, from 76.51 mg H_2_O_2_/g to 38.60 mg H_2_O_2_/g, which is a decrease of 38.60 mg H_2_O_2_/g. It is well known that the Vis/NIR spectra of maize seeds mainly provides chemical information on the main components such as fat, water, and protein, which are related to the extended overtones of C-H, O-H, and N-H, respectively. Differences in the reflectance may be related to the differences in the protein content, which may reflect differences in the maize seed texture parameters. As catalase belongs to the protein group and a different storage time will lead to certain changes in the texture of the corn grain, the difference of the spectral information can be used to detect and identify the corn seeds with a different texture [41]. Therefore, it is necessary to mine the hidden information to further explain the variation characteristics of the texture. Therefore, the hidden information needs to be mined to further explain the variation in the texture properties. Due to the large sample scale, the short sampling interval for ageing, and the inconsistency of the sample placement position affected by temperature and humidity, small fluctuations in the wheat CAT during the ageing process are normal. The CAT of the strong gluten wheat was significantly lower than that of medium gluten and weak gluten wheat in the early stage of ageing, which the texture variation may have caused. Overall, the sensitivity of the wheat CAT was high, and it can be used as an index to characterize the ageing of wheat.

### 2.2. Data Preprocessing

The spectrum extraction process is shown in Figure 2, in which Figure 2a shows the corrected image of the sample on the 80th band after 190 days of storage.

We used the ROI tool in ENVI 5.3 software to select the region of interest, such as the red region in Figure 2b, and select the local rectangular region as the region of interest of the sample, representing the overall information of the sample. Finally, we calculated and saved the average spectral data of the region of interest and drew the average spectral curve using MATLAB R2019a, as shown in Figure 2c.

The average spectral data of all of the samples (420 in total) were extracted by the above method. To reduce these effects of random noise in spectral information and obtain accurate and stable models, we used commonly used preprocessing methods such as standard normal variables (SNV), multiplicative scatter correction (MSC), first derivation (1ST), and the second derivative (2ND) [42]. Figure 3a shows the original average spectra of 420 wheat samples, and Figure 3b shows the spectra after the first-order derivation (1ST) pretreatment. Compared with the original spectra, the band characteristics were enhanced. Figure 3c shows the spectrum after the MSC pretreatment. The spectrum curve was compact, and the effect was outstanding. Figure 3d is the spectrum after SNV preprocessing. The processed spectrum curve was similar to that after MSC processing. The standardization of the spectrum data can eliminate the influence of the data dimension and make the data index comparable, which is convenient for model establishment.

### 2.3. Prediction of CAT Activity Based on the Full Band

Before the model was established, the Kennard–Stone (KS) method was used to divide the sample into a training set and a test set at a 3:1 ratio, which included 315 training set samples and 105 test set samples. The training set was used as modelling data, and the test set was used as prediction data to verify the accuracy of the model [43]. The results of the sample division are shown in Table 1. It can be seen that the divided training set sample CAT range was 37.99–117.78 mg H_2_O_2_/g, and the test set sample CAT range was 37.91–114.24 mg H_2_O_2_/g. The training set sample can include the test set sample, indicating that the sample division effect was good, and this is convenient for the model establishment.

The prediction model of CAT activity in the whole spectral range was established, and the treatment methods of Yu et al. were adopted and relatively improved [44]. CAT combined the original (Y) and the full-band data (256 bands in total) that were preprocessed by 1ST, MSC, and SNV to establish a BP neural network (BPNN) model and a support vector regression (SVR) model. The model determination coefficient (R^2^) and the mean square error (MSE) are shown in Table 2.

The original three preprocessed spectral data sets were combined with the BPNN to establish a full-band CAT prediction model. The determination coefficients of R^2^ values were 0.9562, 0.9485, 0.9610, and 0.9670, and the mean square errors (MSE) were 0.0009, 0.0010, 0.0009, and 0.0010, respectively. The prediction results were good, and the best fit between the CAT measured value and the BPNN’s predicted value is shown in Figure 4a. The original and three preprocessed spectral data sets were combined with the SVR to establish a full-band CAT prediction model. The coefficients of determination of R^2^ values were 0.9569, 0.9689, 0.9635, and 0.9638, and the mean square errors (MSE) were 0.0083, 0.0060, 0.0069, and 0.0069, respectively. The prediction results were good, and the best fit between the CAT measurement value and the SVR prediction value is shown in Figure 4b.

It can be seen that the model established by the SVR method had better performance in predicting CAT than the model established by the BPNN method did, but its mean square error was larger than that of the BPNN method. In summary, the eight full-band prediction models based on the wheat CAT activity were all highly accurate, with the R^2^ values all reaching above 0.9485. Among them, 1ST-SVR was the best full-band prediction model for CAT, with R^2^ = 0.9689.

### 2.4. Prediction of CAT Activity Based on Characteristic Wavelength

Due to a large amount of full-band spectral data and more redundant data, it was necessary to reduce the amount of data, shorten the modelling time, and improve the model’s accuracy [45]. In this paper, the SPA, PCA, and PLSR algorithms are compared. Compared with the other two algorithms, the SPA algorithm was used as a flexible variable selection method to select the optimal variable to solve the collinear problem in the calibration process [46]. The SPA only selects a few columns of data in the original spectral data, but it can summarize the spectral variable information of most of the samples, which can avoid the redundancy and duplication of information to the greatest extent, and at the same time, greatly reduce the number of variables in the process of the model’s establishment, the dimensionality of the data, and the complexity of the model, improve the modeling efficiency and speed, as well as the model’s accuracy and prediction performance [47]. Therefore, in this paper, the PSA algorithm was selected as the best algorithm for screening the wavelengths to extract the characteristic wavelengths associated with the wheat CAT. The extraction results are shown in Figure 5.

Among them, Figure 5a shows the characteristic wavelengths extracted by CAT combined with the original spectral data. A total of 16 characteristic wavelengths were extracted, and the corresponding wavelengths were 853.7 nm, 898.2 nm, 908.5 nm, 915.3 nm, 925.6 nm, 983.4 nm, 1037.7 nm, 1118.7 nm, 1289.3 nm, 1368.9 nm, 1408.5 nm, 1643.6 nm, 1653.3 nm, 1666.3 nm, 1685.7 nm, and 1688.9 nm, respectively. Figure 5b shows the characteristic wavelengths extracted from the spectral data of CAT combined with 1ST. A total of 17 characteristic wavelengths were extracted, and the corresponding wavelengths were 891.4 nm, 932.4 nm, 1098.5 nm, 1101.9 nm, 1105.2 nm, 1209.3 nm, 1269.4 nm, 1272.7 nm, 1359.0 nm, 1461.1 nm, 1467.7 nm, 1470.9 nm, 1474.2 nm, 1552.7 nm, 1555.9 nm, 1562.5 nm, and 1569.0 nm, respectively. Figure 5c shows the characteristic wavelengths extracted from the spectral data after CAT was combined with MSC. A total of 11 characteristic wavelengths were extracted, and the corresponding wavelengths were 918.7 nm, 956.2 nm, 986.8 nm, 1061.4 nm, 1125.4 nm, 1149.0 nm, 1192.6 nm, 1368.9 nm, 1382.1 nm, 1653.3 nm, and 1672.7 nm, respectively. Figure 5d shows the characteristic wavelengths extracted from the spectral data after CAT was combined with SNV, and 14 characteristics were extracted in total. The corresponding wavelengths were 929.0 nm, 949.4 nm, 983.4 nm, 1058.0 nm, 1118.7 nm, 1145.6 nm, 1192.6 nm, 1325.8 nm, 1368.9 nm, 1382.1 nm, 1477.5 nm, 1637.2 nm, 1653.3 nm, and 1672.7 nm, respectively.

The BPNN and SVR prediction models were established based on the wheat CAT combined with the characteristic wavelength spectrum data. The model determination coefficients (R^2^) and mean square errors (MSE) are shown in Table 3.

The original and three preprocessed characteristic band spectral data sets were combined with the BPNN to establish a CAT prediction model. The R^2^ values were 0.9483, 0.9105, 0.9648, and 0.9617, and the mean square error MSE were 0.0009, 0.0008, 0.0010, and 0.0010, respectively. The prediction results were good, and the best fit between the CAT measurement value and the BPNN prediction value is shown in Figure 6a. The original and three preprocessed characteristic band spectral data were combined with SVR to establish a CAT prediction model. The R^2^ values were 0.9538, 0.9347, 0.9664, and 0.9620, and the mean square errors (MSE) were 0.0090, 0.0123, 0.0064, and 0.0071, respectively. The prediction results were good, and the best fit between the CAT measured value and the SVR predicted value is shown in Figure 6b.

In summary, compared with the initial SVR model with the full-band data, the simplified MSC-SPA-SVR model showed similar results, indicating that these 14 wavelengths were suitable for predicting the CAT activity. For the CAT which was combined with the preprocessed characteristic band spectrum data to establish a prediction model, the accuracy of the model established by the SVR method was higher than that of the BPNN method, and the characteristic band spectrum extracted by the MSC method was preprocessed. The accuracy of the model established by the data was higher than that of the other preprocessing methods, the accuracy R^2^ of the eight models established based on the characteristic bands was above 0.9105, and the prediction effect was good. Among them, the MSC-SPA-SVR model had the best prediction effect, with R^2^ = 0.9664. The predicted results of the MSC-SPA-SVR model were consistent with the method reference values that can be found in the references, indicating that the model was accurate, fast, and stable.

The prediction model established by the characteristic waveband was compared with the full waveband model. For the original data and the characteristic band model established by the 1ST and SNV preprocessed spectral data, 16, 17, and 14 characteristic bands were extracted, respectively, accounting for only 6.25%, 6.64%, and 5.47% of the total number of bands. Compared with the full-band model, the accuracy of the model was only reduced by 0.32%, 3.53%, and 0.52%, and the prediction performance was still relatively good. For the characteristic band model established by the spectral data processed by MSC, 11 characteristic bands were extracted, which only accounted for 4.30% of the total number of bands. Compared with the full band model, the accuracy of the model increased by 0.30%, and the prediction performance was better. It can be seen that the number of characteristic bands extracted after MSC processing was smaller, and the model accuracy was higher than that of the full-band model, and the effect was better. Although the model’s accuracy established by the characteristic bands extracted by other processing methods were slightly reduced, the prediction effect was still better. The SPA method extracted the characteristic bands, which greatly reduced the amount of data, removed a large amount of redundant information in the spectral data, shortened the modelling time, and improved the modelling accuracy.

### 2.5. Classification of Different Years of Wheat Based on CAT Activity

For the wheat stored for 0–4 years, a BP neural network (BPNN) classification model and a support vector classification (SVC) model were established based on CAT. The classification statistics results are shown in Table 4.

They were divided into five categories according to the storage life. The classification result of the BPNN is shown in Figure 7a, and the classification accuracy (ACC) was only 67.35%. The classification result of SVC is shown in Figure 7b, and the ACC was only 71.43%.

Among them, the classification error of the wheat harvested from 2017 to 2019, i.e., wheat that was stored for 1–3 years, was relatively large, showing a mixed state, indicating that the CAT activity of the wheat stored for 1–3 years had certain similarities and could not be completely distinguished. Therefore, the wheat harvested in 2017–2018, i.e., the wheat that was stored for 2–3 years, represents one category, and the other durations of wheat storage were each classified into four categories (these categories include 2 to 3 years of wheat storage). The BPNN classification accuracy rate was ACC = 81.63%, which was somewhat improved. The SVC classification accuracy rate was ACC = 69.39%, and the accuracy rate decreased slightly, indicating that the BPNN classification effect was better. The wheat that was harvested in 2018–2019, i.e., the wheat that was stored for 1–2 years, represents one category, and the other durations of wheat storage were each classified into four categories (these categories include 1 to 2 years of wheat storage). The BPNN classification accuracy rate was ACC = 87.76%, and the classification accuracy rate was good. The values for the category that included the wheat that had been stored for 2 to 3 years increased. The SVC classification accuracy rate was ACC = 87.76%. The accuracy rate was good, and it had increased. Therefore, this indicates that the CAT activity of the wheat that had been stored for 1–2 years was more mixed and difficult to distinguish. The wheat harvested in 2017–2019, i.e., the wheat that was stored for 1–3 years represents one category, and the other durations of wheat storage were each placed into this category to make a total of three classifications. The BPNN classification accuracy rate was ACC = 100%, as shown in Figure 8a. The SVC classification accuracy rate was ACC = 95.92%, as shown in Figure 8b.

The accuracies of the two classification methods were both optimal, and the BPNN classification method can completely distinguish the three types of wheat. By taking the wheat harvested in 2017–2019 that was stored for 1–4 years as the first category, the newly harvested wheat and the stock wheat were classified into two categories. The classification accuracy of the BPNN was ACC = 100%, and the accuracy of the SVC classification was ACC = 95.92%. The classification effects were better, and the BPNN method’s classification effect was better.

Based on the above classification results, the classification model with the highest classification accuracy and the most categories (i.e., the BPNN had three classifications) was selected to classify the wheat ageing. The classification model was tested multiple times to obtain the classification threshold on CAT. The ageing of the newly harvested wheat was of grade Ⅰ, and the critical value of CAT was 71.42 mg H_2_O_2_/g. The ageing of the wheat that had been stored for 1–3 years was of grade Ⅱ, and the critical value of CAT was 37.91 mg H_2_O_2_/g. The wheat that had been stored for more than three years was of grade Ⅲ. Namely, when the content of CAT was more than 71.42 mg H_2_O_2_/g, the wheat was fresh when it was in grade Ⅰ. When the content of CAT ranged from 71.42 mg H_2_O_2_/g to 37.91 mg H_2_O_2_/g, the wheat was fresh when it was in grade Ⅱ. When the content of CAT was less than 37.91 mg H_2_O_2_/g, the wheat was fresh when it was in grade Ⅲ.

### 2.6. Visualization of Chemical Information

Since the hyperspectral image of a wheat sample contains a large number of pixels, up to more than 80,000, here, only a 100 × 100 square area (a total of 10,000 pixels) was selected for the spectral data extraction and prediction. We visualized the chemical information distribution of the wheat CAT activity, and we used the optimal CAT prediction model (MSC-SPA-SVR) which we have established to predict the CAT of each pixel, which was applied to transform and visualize every pixel of the hyperspectral images into the corresponding color images to predict the CAT activity distribution of the tested wheat. The obtained visualized image was displayed in a linear color scale with different colors, reflecting the changes of the CAT activity. All of the procedures of the visualization process were carried out with the software Matlab 2020a (The MathWorks Inc., Natick, MA, USA). Figure 9 shows the pseudocolor images of the CAT visualization distribution of three wheat samples with different ageing grades.

The color of the image changed from blue to green and from yellow to red. The bluer the color is, then the higher the CAT value is, which means that the wheat is newer. The redder the color is, the lower the CAT value is, indicating that the wheat is older. As shown in Figure 9a, when the CAT = 112.78 mg H_2_O_2_/g, and the wheat is of grade Ⅰ and is fresh, the overall image is blue-green, and yellow and red colors are slightly distributed. As shown in Figure 9b, the image is yellow when the CAT = 51.54 mg H_2_O_2_/g, and the wheat is of grade Ⅱ and is fresh. The amount of red is obviously increased, and it is distributed with blue-green. As shown in Figure 9c, when the CAT = 37.91 mg H_2_O_2_/g, and the wheat is of grade Ⅲ and is fresh, the overall image is yellow-red, and blue-green distributed in a small amount. However, it is easy and convenient to predict and visualize the CAT activity of various parts of the wheat through the HSI. The HSI described the spatial and spectral data as a complete whole. Therefore, the HSI imaging technology has the superpotential to quickly, non-destructively, and visually measure the activity of the wheat CAT, which can further discriminate the ageing of the wheat.

To sum up, the HSI combined with the BPNN and SVM was successfully used to predict the wheat’s CAT activity and characterize the wheat’s ageing. In the artificially accelerated ageing process, the CAT activity of the strong, medium, and weak gluten wheat decreased with the extension of the ageing time. The wheat CAT activity and ageing time were more sensitive, and they could be used as an indicator that characterizes the ageing of the wheat. Using the 1ST, MSC, and SNV methods to preprocess the wheat hyperspectral data can not only eliminate part of the noise, but they can also strengthen the spectral information, and the effect was significantly improved. The CAT activity was combined with the original and full-band spectral data which had been preprocessed by 1ST, MSC and SNV to establish the BPNN model and SVR model. The prediction effect of the model was significant. All of them are above 0.9485, among which the 1ST-SVR model had the best prediction effect (R^2^ = 0.9689). Using SPA to extract the characteristic bands of the wheat CAT activity and modelling, there were fewer characteristic bands extracted after the MSC pretreatment (11) (only 4.30% of the total wavelengths) and the corresponding wavelengths were 918.7 nm, 956.2 nm, 986.8 nm, 1061.4 nm, 1125.4 nm, 1149.0 nm, 1192.6 nm, 1368.9 nm, 1382.1 nm, 1653.3 nm, and 1672.7 nm. The MSC-SPA-SVR model had the best prediction effect (R^2^ = 0.9664). The MSC pretreatment method performed better, and the SVR modelling method performed better. Furthermore, compared with other studies of the CAT activity by spectroscopy [48,49], the CAT activity prediction model in the present study had a better performance. Therefore, the results illustrated that the CAT activity prediction model based on the hyperspectral image can realize the fast and nondestructive CAT activity detecting of maize kernels.

## 3. Materials and Methods

### 3.1. Sample Processing

Newly harvested wheat: The newly harvested wheat 26 (strong gluten) and the newly harvested wheat 35 (medium gluten) were from the Xinxiang Academy of Agricultural Sciences; Yang wheat 13 (weak gluten) was from the Xinyang Academy of Agricultural Sciences; both of them were newly harvested in June 2020. After the cleaning and impurity removal, the wheat was sub-packed in non-woven bags. Each type of wheat was divided into 140 portions, each portion was 150 g, and there were 420 samples in total. For accelerated ageing, the wheat samples were placed in a constant temperature and humidity incubator. The storage temperature was set at 40 ± 1 °C, and the relative humidity was set at 90 ± 5% RH. The samples were sampled every ten days for the measurement.

Stored wheat: A total of 147 pieces of wheat of different ages were collected in Henan Province, all of which were mixed wheat. There were 35 wheat samples which were harvested in 2016, 32 wheat samples which were harvested in 2017, 25 wheat samples which were harvested in 2018, 34 wheat samples which were harvested in 2019, and 21 wheat samples which were harvested in 2020. Each wheat sample was 150 g.

### 3.2. Determination of CAT

The leaf H_2_O_2_ contents and CAT activities were measured on the third leaves of the control and drought-treated wheat plants. H_2_O_2_ was determined using the method [50]. The CAT was assayed as described [51].

### 3.3. Hyperspectral Image Acquisition and Correction

This study used the Gaia Sorter-Dual full-band hyperspectral imaging system, which was produced by Sichuan Shuangli Hepu Technology Co., Ltd., as shown in Figure 10.

Its core components include a bromine tungsten light source, a spectroscopic camera (Image-λ-N17E), an electronically controlled mobile platform, a computer, and control software. The spectral range of the system was 850–1700 nm, the number of spectral bands was 256. The pure genotypes of soft red winter wheat and soft white wheat were placed in a Petri dish for hyperspectral image scanning. In order to obtain a clear and undistorted image, the system parameters were determined after multiple scans [52]. The camera height was 13.5 cm, the exposure time was 42 ms, and the platform movement speed was 0.48 cm/s. In order to reduce the influence of light source changes and system noise, the obtained original image (R0) was subjected to black and white correction. A blackboard image (B), with a reflectivity close to 0%, was obtained by covering the lens with a lens cap. A whiteboard image (W), with a reflectivity close to 100%, was obtained from a PTFE sheet. The corrected image (RC) was calculated according to Formula (1).
(1)$$R_{C}=\frac{R_{0}-B}{W-B}\times 100\%$$

### 3.4. Data Analysis

The ROI tool in the ENVI 5.3 software (ENVI 5.3, Research Systems Inc., Solutions, Boulder, CO, USA, 2014) was used to select the region of interest (ROI) of the corrected hyperspectral images and to calculate and extract the average spectral data of the region of interest (ROI). MATLAB R2019a software (R2019a, The MathWorks Inc., MA, USA) was used to process and model the extracted spectral data. In this study, the first derivative (1ST), the multiple scattering correction (MSC), and the standard normal variable transformation (SNV) methods were used to preprocess 420 extracted spectral data to eliminate the scattering effects caused by system noise and varying particle size and distribution [53,54]. The successive projections algorithm (SPA) was used to extract the characteristic wavelengths related to the wheat fatty acid values and bean plant CAT activity, and the BP neural network (BPNN) and support vector machine (SVM) methods were used to establish the regression and classification models, respectively [55,56].

### 3.5. Visualization of Chemical Information

After the correction of all of the hyperspectral image pixels were extracted from the spectral data, and the optimal prediction model was imported into the chemical information of each pixel in the forecast, and the forecast data in the original coordinate values of the pixels were used to form a two-dimensional matrix, the pseudocolor processing was conducted to obtain the distribution of the characterization of chemical information pseudocolor image, and the distribution of chemical information visualization was realized [57].

## 4. Conclusions

Hyperspectral imaging technology (850–1700 nm), combined with the BPNN and SVM machine learning method, successfully predicted the CAT activity of wheat and characterized the aging of wheat. The experimental results show that the MSC-SPA-SVR model is the best in quantitatively predicting the wheat CAT activity. Based on CAT to classify the wheat of different ages, the accuracy of the BPNN’s three classifications was 100%. Moreover, the best obtained MSC-SPA-SVR model was used to transfer the spectrum of each pixel into its wheat CAT activity values, thus, visualization maps of the CAT activity distribution were generated. The results of this study indicate that the HSI has a good predictive ability and potential for wheat CAT activity. As far as it is known, it is the first time that researchers have used hyperspectral imaging technology to predict the activity of wheat catalase. The variety and quantity of wheat used in the experiment are limited, but the experimental results show that the experimental method is feasible, and this technology can be extended to more varieties of wheat, wheat samples, or other grain reserves, which will make the country’s detection and monitoring of grain reserves more rapid, convenient, and non-destructive.

## Figures and Tables

**Figure 1 molecules-27-08648-f001:**
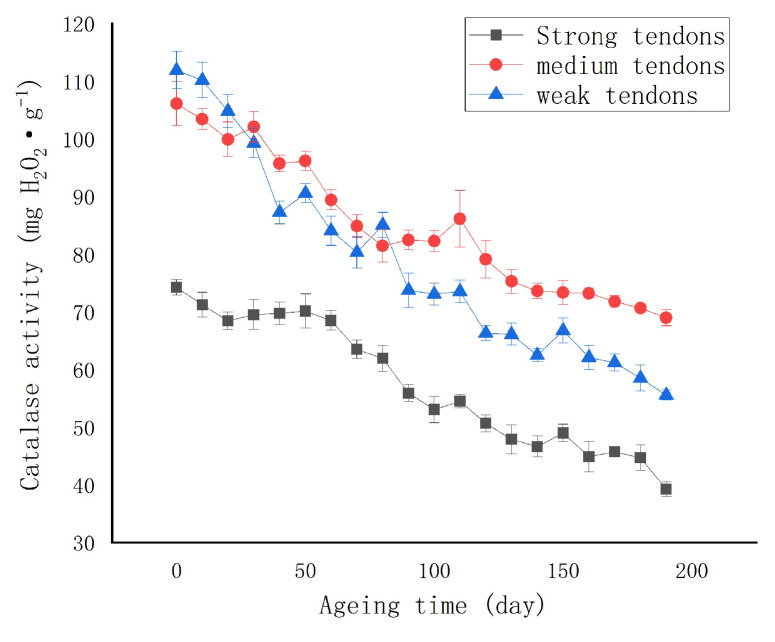
Changes in catalase activity during ageing.

**Figure 2 molecules-27-08648-f002:**
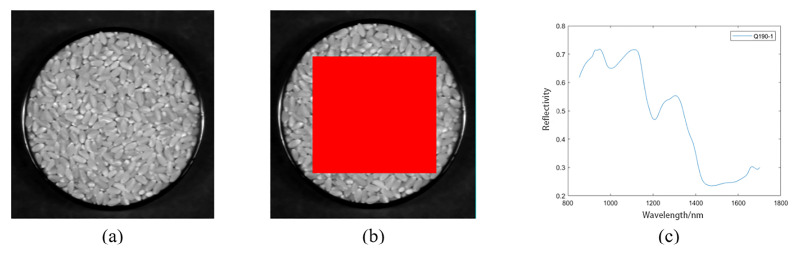
Extracting process of spectral data. (**a**) the corrected image of the sample; (**b**) the selected region of interest; (**c**) the average spectral curve of the region of interest.

**Figure 3 molecules-27-08648-f003:**
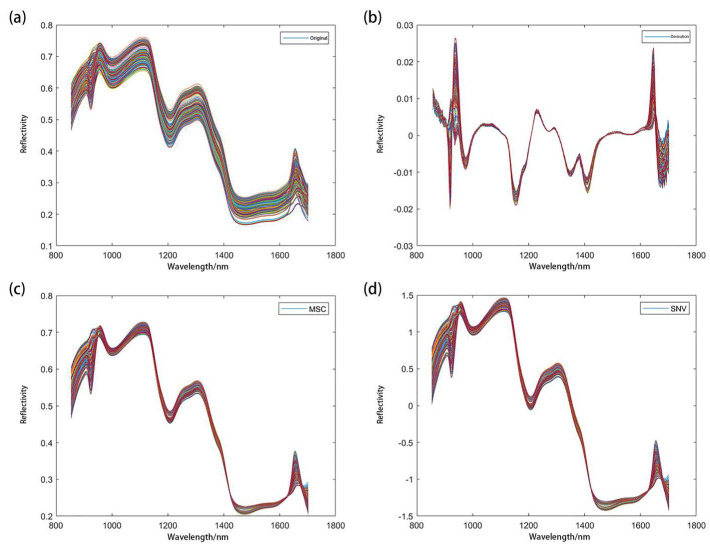
The spectra of original and preprocessed data.

**Figure 4 molecules-27-08648-f004:**
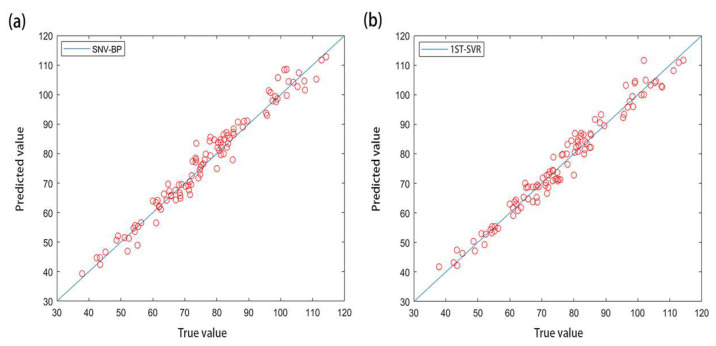
(**a**,**b**) represent the fitting results of the SNV-BPNN model and 1ST-SVR model based on full frequency band, respectively.

**Figure 5 molecules-27-08648-f005:**
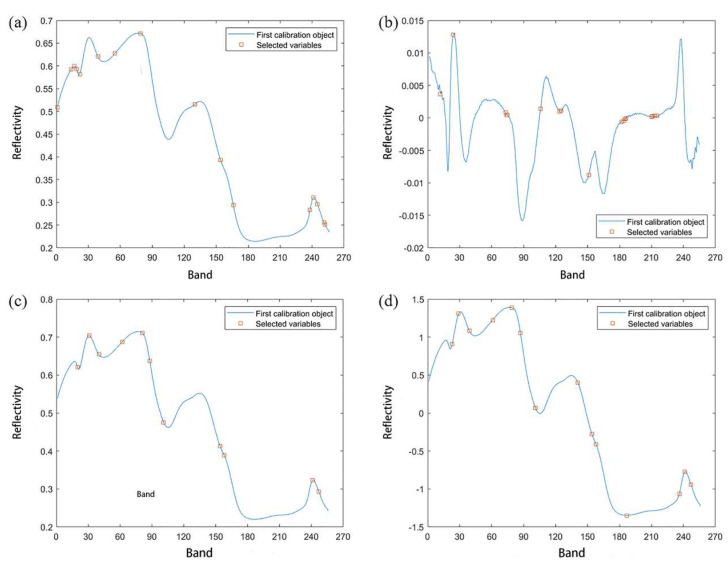
Extracting results of the characteristic wavelength.

**Figure 6 molecules-27-08648-f006:**
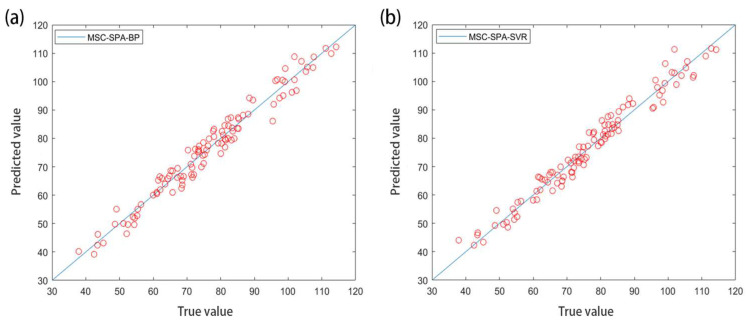
(**a**,**b**) represent the fitting results of MSC-SPA-BPNN and MSC-SPA-SVR models based on feature bands, respectively.

**Figure 7 molecules-27-08648-f007:**
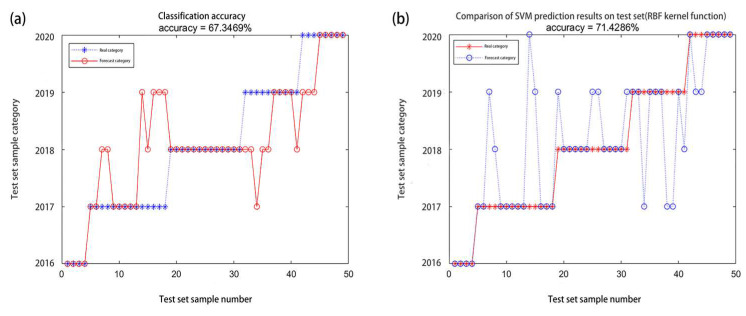
(**a**,**b**) represent the five categories results of catalase activity BPNN and SVC, respectively.

**Figure 8 molecules-27-08648-f008:**
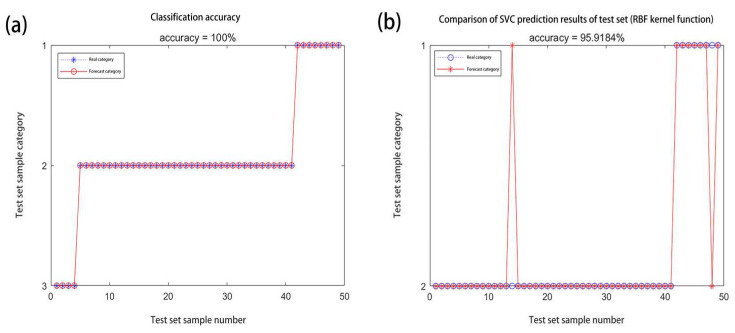
(**a**,**b**) represent the classification results of BPNN and SVC, respectively.

**Figure 9 molecules-27-08648-f009:**
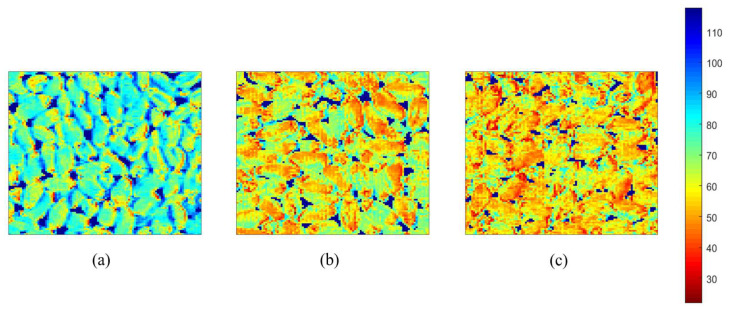
Visible distribution of catalase activity.

**Figure 10 molecules-27-08648-f010:**
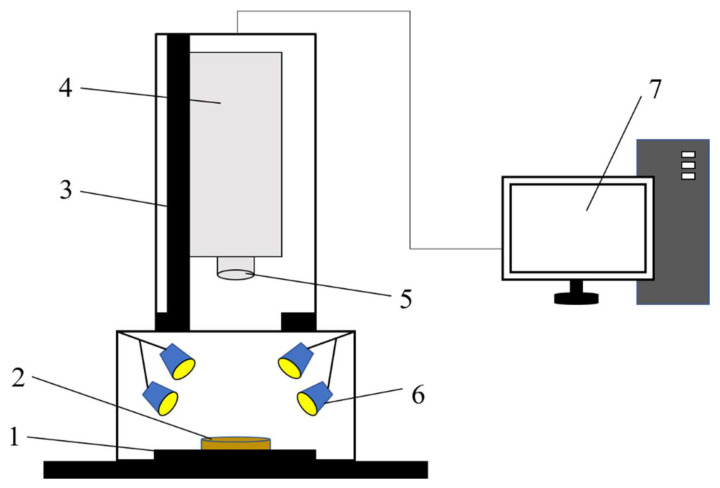
Schematic diagram of a hyperspectral imaging system. 1. Electronically controlled mobile platform; 2. Sample; 3. Camera lifting platform; 4. Hyperspectral imager; 5. Lens; 6. Light source; 7. Computer.

**Table 1 molecules-27-08648-t001:** Dividing results of sample.

Data Set	Sample Size	CAT Activity (mg H_2_O_2_/g)			
		Maximum	Minimum	Average Value	Standard Deviation
Training set	315	117.78	37.99	72.66	18.14
Test set	105	114.24	37.91	76.49	17.333

**Table 2 molecules-27-08648-t002:** Modeling results based on the full band.

Pretreatment	Neural Networks (BP)		Support Vector Regression (SVR)	
	R^2^	Mean Square Error (MSE)	R^2^	Mean Square Error (MSE)
Original (Y)	0.9562	0.0009	0.9569	0.0083
1ST	0.9485	0.0010	0.9689	0.0060
MSC	0.9610	0.0009	0.9635	0.0069
SNV	0.9670	0.0010	0.9638	0.0069

**Table 3 molecules-27-08648-t003:** Modeling results based on the characteristic band.

Pretreatment	Neural Networks (BP)		Support Vector Regression (SVR)	
	R^2^	Mean Square Error (MSE)	R^2^	Mean Square Error (MSE)
Original (Y)	0.9483	0.0009	0.9538	0.0090
1ST	0.9105	0.0008	0.9347	0.0123
MSC	0.9648	0.0010	0.9664	0.0064
SNV	0.9617	0.0010	0.9620	0.0071

**Table 4 molecules-27-08648-t004:** Classifying results based on catalase activity.

Category	BPNN Classification		Support Vector Classification (SVC)	
	Correct Rate/%	Number of Correct Classifications	Correct Rate/%	Number of Correct Classifications
Five categories	67.35	33	71.43	35
Four categories: 2 to 3 years of storage	81.63	40	69.39	34
Four categories: 1 to 2 years of storage	87.76	43	87.76	43
Three categories	100	49	95.92	47
Two categories	100	49	95.92	47

## Data Availability

The data presented in this study are available on request from the corresponding author.

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
