# Peer review of "Non-Destructive Hyperspectral Imaging for Rapid Determination of Catalase Activity and Ageing Visualization of Wheat Stored for Different Durations"

_molecules, 2022, doi:10.3390/molecules27248648_

Round 1
Reviewer 1 Report
This is an excellent study and of great interest to the science community. The application of hyperspectral imaging and development of algorithms is novel.
Overall the grammar is very good, however there are parts of the manuscript can be grammatically improved
The Conclusion is long and some paragraphs should be in the Discussion..
The authors mention pollution-free technology, but it is not clear what this refers to?
Author Response
Response to Reviewer 1
Point 1: Overall the grammar is very good, however there are parts of the manuscript can be grammatically improved.
Response 1: Thank you for your nice comment. The authors checked the whole manuscript again. Some parts of the manuscript have been improved grammatically.
Point 2: The Conclusion is long and some paragraphs should be in the Discussion.
Response 2: Thank you for your nice comment. The authors separated parts of the conclusions, added them to the discussion, and marked them in yellow (Line 432-Line446).
Point 3: The authors mention pollution-free technology, but it is not clear what this refers to?
Response 3: Thank you for your nice comment. The authors are sorry to bring these ambiguities to the reviewers. It means that we can build a quantitative predictive model for the classification of peroxidase activity of wheat aging in different years by hyperspectral imaging combined with the corresponding BPNN SVM to achieve the detection of wheat without changing the basic structure of wheat and without applying chemical reagents. In this paper, the non-pollution testing technology has been changed into the non-destructive testing technology, and marked with yellow (Line 19 and Line 446- ‘non-destructively technology’ instead of ‘pollution-free technology’).
Reviewer 2 Report
In this study, hyperspectral imaging technology combined with BPNN and SVM was used to establish a quantitative prediction model for catalase activity of wheat with the classification of the ageing of wheat based on different years. This study was valuable work for application. This paper was well-written. The author should address the following minor comments before publication.
1) All Figures should be optimized. They are not clear to figure out, perhaps the problem of resolution ratio.
2) In Fig. 3c, there are some Chinese characters. Please pay attention and be serious.
3) In Fig. 4, there is no introduction of each sub-figure. The author should give some brief information.
4) In Fig. 5b, what was the ‘DQ-SVR’? DQ was used as the Pretreatment method?
5) Why did the author use SPA as the wavelength selection method to extract characteristic wavelengths? There are a lot of wavelength selection methods introduced in some review papers.
6) Line 245, continuous projection algorithm (SPA), sure?
7) The author cited 56 references. However, the number of references in recent 3 years is relatively low.
Author Response
Response to Reviewer 2
Point 1: All Figures should be optimized. They are not clear to figure out, perhaps the problem of resolution ratio.
Response 1: Thank you for your nice comment. It has been changed according to your requirements (Figure 1-Figure 10).
Point 2: In Fig. 3c, there are some Chinese characters. Please pay attention and be serious.
Response 2: Thank you for your nice comment. The Chinese characters in Figure 3c has been changed to English (Figure 3c).
Point 3: In Fig. 4, there is no introduction of each sub-figure. The author should give some brief information.
Response3: Thank you for your nice comment. The sub-figure information has been introduced (Line 126-Line 136).
Point 4: In Fig. 5b, what was the ‘DQ-SVR’? DQ was used as the Pretreatment method?
Response 4: Thank you for your nice comment. The QD in QD-SVR in Figure 5b means the first derivative method in the derivative method of the pretreatment method, and the QD-SVR has been modified to 1ST-SVR in Figure 5b (Figure 5b).
Point 5: Why did the author use SPA as the wavelength selection method to extract characteristic wavelengths? There are a lot of wavelength selection methods introduced in some review papers.
Response 5: Thank you for your nice comment. In this paper, SPA, PCA, and PLSR algorithms are compared. Compared with the other two algorithms, the SPA algorithm is used as a flexible variable selection method to select the optimal variable to solve the collinear problem in the calibration process. SPA only selects a few columns of data in the original spectral data, but can summarize the spectral variable information of most samples, which can avoid the redundancy and duplication of information to the greatest extent, and at the same time, greatly reduce the number of variables in the process of model establishment, the dimensionality of the data and the complexity of the model, improve the modeling efficiency and speed, as well as the model accuracy and prediction performance. Therefore, the PSA algorithm was selected as the best algorithm for screening wavelengths in this article. And the explanation is given in the corresponding place in the article, and the explanation part is marked in yellow (Line 183-Line 193).
Point 6: Line 245, continuous projection algorithm (SPA), sure?
Response 6: the authors revised it. Line 418.
Point 7: The author cited 56 references. However, the number of references in recent 3 years is relatively low.
Response 7: Thank you for your nice comment. Appropriately increase the number of references cited in the past three years (references1、references5、references39、references50、references51).